# Plasmon mediated coherent population oscillations in molecular aggregates

Daniel Timmer [1,8], Moritz Gittinger [1,8], Thomas Quenzel[1], Sven Stephan[1,6], Yu Zhang [2], Marvin F. Schumacher [3], Arne Lützen [3], Martin Silies[1,6], Sergei Tretiak [2], Jin-Hui Zhong [1,7], Antonietta De Sio [1,4] & Christoph Lienau [1,4,5] ✉

The strong coherent coupling of quantum emitters to vacuum fluctuations of the light field offers opportunities for manipulating the optical and transport properties of nanomaterials, with potential applications ranging from ultra-sensitive all-optical switching to creating polariton condensates. Often, ubiquitous decoherence processes at ambient conditions limit these couplings to such short time scales that the quantum dynamics of the interacting system remains elusive. Prominent examples are strongly coupled exciton-plasmon systems, which, so far, have mostly been investigated by linear optical spectroscopy. Here, we use ultrafast two-dimensional electronic spectroscopy to probe the quantum dynamics of J-aggregate excitons collectively coupled to the spatially structured plasmonic fields of a gold nanoslit array. We observe rich coherent Rabi oscillation dynamics reflecting a plasmon-driven coherent exciton population transfer over mesoscopic distances at room temperature. This opens up new opportunities to manipulate the coherent transport of matter excitations by coupling to vacuum fields.

The strong coupling of quantum emitters to light[1,2] emerges as a critical instrument for directing the optical[3,4] and electronic transport[5–8] properties of nanomaterials by all-optical means[3]. These phenomena can be exploited to modify the outcome of photochemical reactions in the electronic ground[9] and excited states[7,10–12], for creating new and unusual states of condensed matter systems such as polariton condensates[13] at room temperature[14] or for designing entirely new optical device concepts such as polaritonic lasers[15] operating at room temperature[16]. In particular the strong coupling of quantum emitters to surface plasmon excitations in metallic nanostructures[17–20] has caught much attention since the strong nanometer-scale spatial confinement of the plasmonic mode[21–24] offers a direct path for locally enhancing vacuum field fluctuations[25] and—thus—the coupling to

quantum matter. Remarkably, it has led to the demonstration of strong coupling to a single molecule[26] or quantum dot[19,27], fundamental hallmarks of quantum plasmonics[28].

Optical excitations in metallic nanostructures are inherently short lived due to substantial Ohmic and radiative losses[22]. Consequently, quantum coherence in metal-based hybrid systems is usually lost after a few tens to hundreds of femtoseconds[21]. This has, so far, limited experimental work to mostly linear optical studies, either on ensembles or single quantum structures. Advanced two-dimensional coherent electronic spectroscopies (2DES)[29] have emerged as powerful tools for probing quantum-coherent couplings in the time domain[30–32] even in strongly dephasing media[33,34] and particularly for accessing many-body excitations[35–38] in quantum systems. However, these techniques

[1]Institut für Physik, Carl von Ossietzky Universität, Oldenburg, Germany. [2]Theoretical Division, Los Alamos National Laboratory, Los Alamos, NM, USA. [3]Kekulé-Institute for Organic Chemistry and Biochemistry, University of Bonn, Bonn, Germany. [4]Center for Nanoscale Dynamics (CeNaD), Carl von Ossietzky Universität, Oldenburg, Germany. [5]Forschungszentrum Neurosensorik, Carl von Ossietzky Universität, Oldenburg, Germany. [6]Present address: Institute for Lasers and Optics, University of Applied Sciences, Emden, Germany. [7]Present address: Department of Materials Science and Engineering, Southern University of Science and Technology, Guangdong, China. [8]These authors contributed equally: Daniel Timmer, Moritz Gittinger. ✉e-mail: christoph.lienau@uni-oldenburg.de

are yet to be applied to hybrid plasmonic systems in the strong coupling regime. In 2DES, the excitation with a pair of phase-locked, short optical pulses allows for the selective excitation of different resonances in the system. A third, time-delayed probe pulse generates two-dimensional energy-energy maps of the optical response that correlate the optically excited and detected resonances. Distinct cross peaks in these maps, oscillating as a function of the time delay between excitation pair and optical probe, are unambiguous signatures of strong, quantum-coherent couplings in the hybrid systems.

Here, we demonstrate such temporally oscillating cross peaks in the 2DES spectra of a model molecular aggregate system revealing collective strong coupling between excitons and surface plasmon polaritons. Our results show that these oscillations are dominated by a robust coherent population transfer between spatially separated strongly and weakly coupled excitons in different regions of the nanostructure, mediated by the plasmonic field. This suggests light-driven coherent transport of matter excitations in nanosystems over mesoscopic distances at ambient conditions.

## Results and discussion

To probe strong couplings between excitons and plasmons, we performed angle-resolved 2DES on a hybrid plasmonic cavity, a gold nanoslit array covered with a J-aggregated thin film (Fig. 1a). This molecular aggregate is based on squaraine monomers (ProSQ-C16, inset in Fig. 1b). Their electronic properties are reasonably well described within phenomenological essential state models[39], showing that only the lowest-lying electronically excited state is relevant for the present work. When deposited on gold, they form well-ordered

J-aggregated thin films[40,41]. Dipolar coupling among neighboring molecules results in a delocalization of the optical excitation across ~20–30 monomers at room temperature and in the formation of strongly red-shifted and spectrally narrow superradiant exciton $|X\rangle$ resonances at around 1.59 eV[41] (Fig. 1b). In 2DES, this results in a spectrally sharp and well isolated exciton peak with a dispersive line shape along the detection axis (Fig. 1c). Such 2DES maps are recorded by exciting the sample with a pair of collinear pump pulses with time delay $\tau$ and by probing the pump-induced change in sample reflectivity with a broadband probe pulse that is delayed by a waiting time $T$ with respect to the second pump. Fourier transform of the resulting differential reflectivity spectra recorded for various $\tau$, at fixed $T$, results in the 2DES map in Fig. 1c (see Methods). In the following, we label the peaks in these maps as (ex,det), where ex and det denote the excited and detected resonances, respectively. The dispersive line shape arises from a superposition of three contributions: ground state bleaching (GSB) and stimulated emission (SE) of the one-quantum $|X\rangle$ resonance overlap with an excited state absorption (ESA) from $|X\rangle$ to two-exciton, $|XX\rangle$, states. In J-aggregates, the two-quantum $|XX\rangle$ resonances are blue-shifted ($\Delta E$) since Pauli-blocking in each monomer dictates that the lowest-lying delocalized exciton state in every aggregate can be populated only once[42]. Slight line broadenings result from higher-lying aggregated exciton states. Exciton relaxation within the disordered aggregates leads to a partial decay of the $|X\rangle$ peak on a 100-fs time scale (Fig. 1d).

We deposit such J-aggregated thin films, with a thickness of 10 nm, onto a plasmonic nanoslit array, milled into a 200-nm thick gold film. The nanoslits form a cavity that locally confines optical near fields and

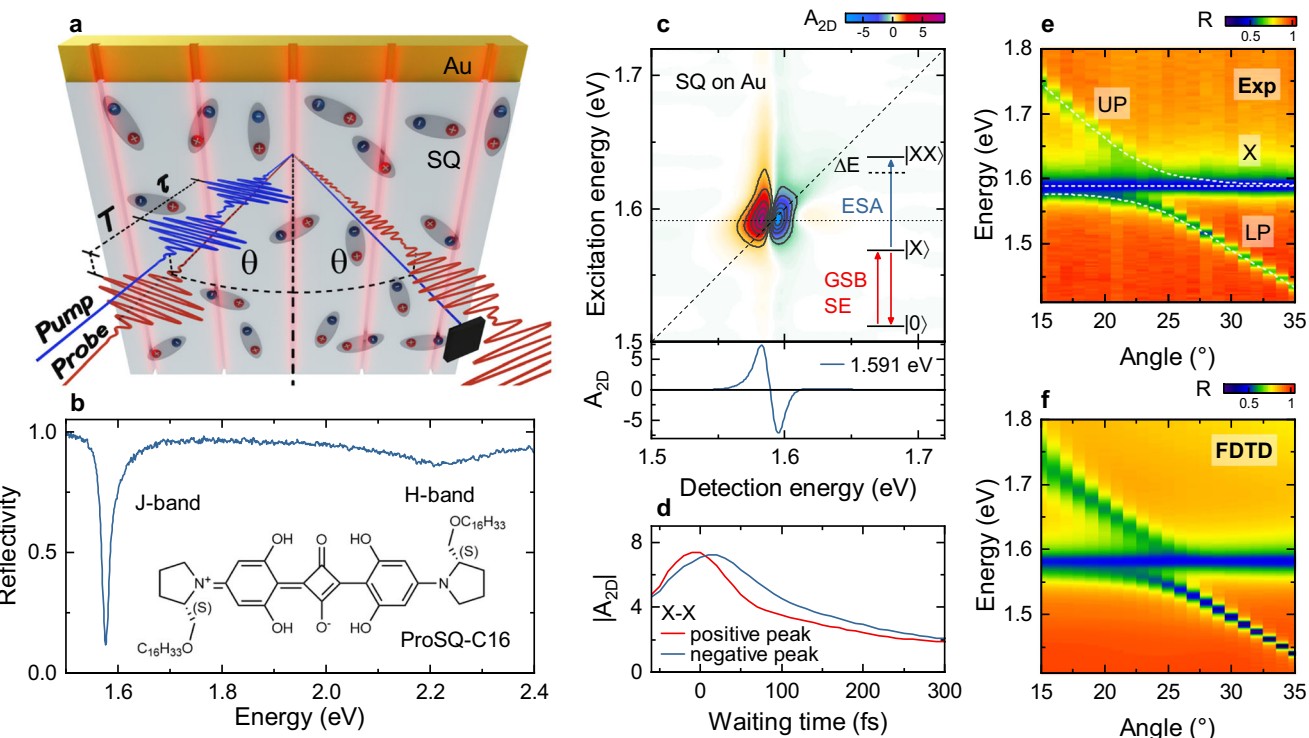

**Fig. 1 | Strong coupling of a gold (Au) nanoslit array coated with a squaraine-based J-aggregated thin film (ProSQ-C16, SQ). a** Experimental geometry. The nanoslit sample is illuminated with a phase-locked pair of pump pulses, separated by the coherence time T at incidence angle $\theta$. The pump-induced change in sample reflectivity is monitored by a probe pulse with the same incidence angle and time-delayed by the waiting time T. **b** Linear reflectivity of a 10-nm J-aggregated film of ProSQ-C16 squaraine molecules (inset) on a gold surface. **c** Experimental, reflective two-dimensional electronic (2DES) spectrum of the film on a flat gold surface at $T = 0$ fs. Bottom: Cross section along the detection energy for excitation at 1.591 eV

showing the characteristic dispersive line shape of the J-aggregate exciton. Inset: One-quantum ($|X\rangle$) and two-quantum ($|XX\rangle$) excitations contributing to the 2DES exciton peak. **d** Waiting time dynamics at the maximum (red) and minimum (blue) of the 2DES exciton peak, showing incoherent relaxation dynamics on a 100-fs time scale. **e** Angle-resolved linear reflectivity spectra reveal the dispersion relations of upper (UP) and lower (LP) polaritons together with an angle-independent peak of "uncoupled" excitons (X). **f** Finite-difference time domain (FDTD) simulation of the angle-resolved reflectivity.

strongly enhances their coupling to the in-plane component of the exciton dipole moment (Fig. S11). Width, height (45 nm) and period (530 nm) of the array are chosen to create sharp surface plasmon polariton (SPP) resonances of the grating with an energy that can be tuned across the exciton resonance by varying the incidence angle $\theta$ (Fig. S1a). Angle-resolved linear reflectivity spectra (Fig. 1e) show that the collective dipolar coupling between excitons and SPPs results in the formation of mixed upper (UP) and lower (LP) polariton branches. From the avoided crossing, we deduce a normal mode splitting of ~60 meV, twice the Rabi energy $\hbar\Omega_R$. The polariton branches are superimposed by an angle-independent exciton peak. It is commonly thought to arise from "uncoupled" excitons which are only weakly interacting with the SPP field, e.g., because they lie in regions outside the slits with much reduced local field enhancement[21,43,44]. This interpretation of the linear spectra is well supported by finite difference time domain (FDTD) simulations of Maxwell's equations (Fig. 1f and S1b).

Most of these resonances also appear in angle-dependent 2DES spectra recorded at $T = 0$ fs (Fig. 2a–c and Fig. S3). Along their diagonal, we observe strong (LP,LP) and (X,X) peaks with dispersive line shapes along $E_{det}$. In contrast, the (UP,UP) peak is much weaker and appears only at angles below the crossing at $\theta_c = 23°$. In addition, we find pronounced cross peaks between LP and X, both below and above the diagonal. Their dispersive line shapes are best seen for $\theta = 27°$ in Fig. 2c. Cross peaks between UP and both, LP and X, are much weaker in amplitude and are only resolved for $\theta < \theta_c$. While (X,UP) and (UP,X) have dispersive line shapes, the other weak UP peaks appear absorptive in shape. Resonance energies and spectral line shapes deduced from 2DES are supported by angle-resolved pump-probe measurements, shown in Fig. S5. The quite pronounced cross peaks between "uncoupled" excitons and polaritons are unexpected since the uncoupled excitons are thought to be spatially well separated and thus essentially uncorrelated with those excitons that are hybridized with the SPP field. Hence, it is not obvious that their excitation should result in a polariton nonlinearity.

The observation of dispersive line shapes for both diagonal and cross peaks now allows us to correlate one-quantum (1Q) resonances, characterized by a positive GSB and SE peak, and two-quantum (2Q) resonances with a negative ESA signal[41,45]. We deduce 1Q energies from peak maxima along $E_{ex}$, while 2Q energies are taken as the zero crossing of a dispersive peak along $E_{det}$. The resulting energies are

plotted in Fig. 2d as open circles, together with the 1Q dispersion (solid lines) deduced from angle-resolved reflectivity. In addition, the 2Q dispersions are estimated by adding the energies of the contributing 1Q states without further corrections. Obviously, the 1Q energies obtained from 2DES match those deduced from linear spectroscopy, while the 2Q dispersions show several new features. Since the experiment probes the collective coupling of many excitons to a single plasmonic mode, we expect, from a commonly employed Tavis Cummings (TC) model, to observe three distinct 2Q states[35,45]. The model predicts doubly excited 2LP and 2UP polaritons and a mixed UP/LP state, while all other states remain optically inactive ("dark")[45]. Indeed, these resonances have been seen in the 2Q dispersions measured for semiconductor microcavities[35] and a TC model has recently also been used to discuss organic microcavity polaritons[45]. As a result of the fermionic nonlinearity induced by the exciton part of the wavefunction, the energies of the 2Q states are slightly blue-shifted with respect to twice the 1Q transition. This blue shift is proportional to the two-exciton fraction of the 2Q wavefunction[35,45]. Since the doubly excited X state is uncoupled from the plasmon branch, an angle-independent XX contribution is expected (Fig. 2).

In the experiments, the 2LP resonances are clearly resolved, while 2UP and UP/LP are apparently lacking. The mixed resonances (gray and black circles) follow the UP/X and LP/X dispersions with a distinct avoided crossing at $\theta_c$. The appearance of those resonances goes beyond the TC model and requires further discussion.

Before that, however, we inspect the waiting time dynamics of the 2DES spectra. This is exemplarily done in Fig. 3a for $\theta = 27°$. Indeed, we observe pronounced coherent oscillations of the amplitude on all diagonal and cross peaks in the 2DES map, except for the (X,X) peak. We will argue below that the oscillatory waiting time dynamics of the 2DES peaks provides crucial new information about coherent couplings in exciton-plasmon systems that has not been obtained from one-dimensional pump-probe spectroscopy, as previously reported by some of the present authors[21]. Interestingly, for $\theta = 27°$ (Fig. 3a), the oscillation period $T_X = 2\pi/(\omega_X - \omega_{LP})$ matches the splitting between the X and LP resonances. Coherent oscillations with the same period are also observed in pump-probe spectra (Fig. 3e, Fig. S5). The traditional understanding of strong X-SPP coupling would, instead, predict Rabi oscillations with a period $T_R = 2\pi/(\omega_{UP} - \omega_{LP})$ given by the normal mode splitting of the polaritons[1,21,22,46]. Angle-dependent pump-probe transients, detected at the LP resonance (Fig. 3e and Fig. S5),

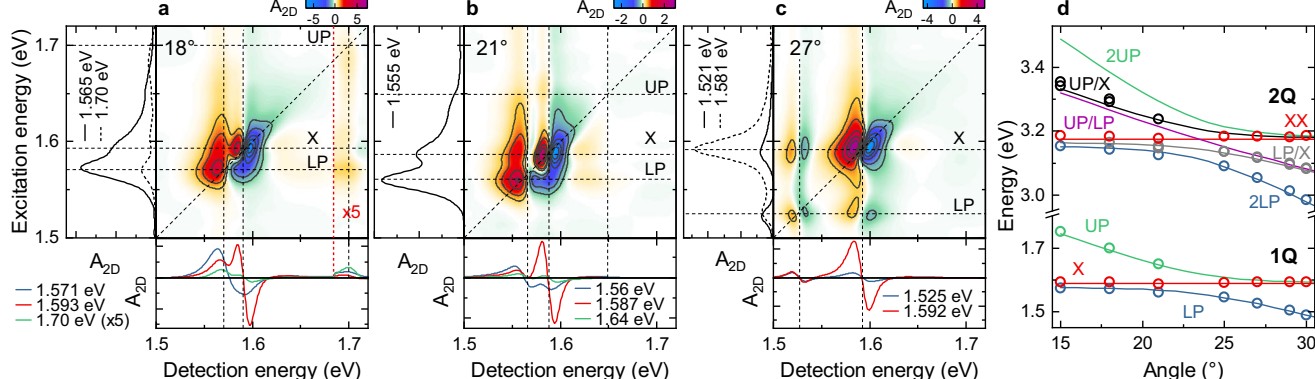

**Fig. 2 | Experimental 2DES maps of the J-aggregate-nanoslit array for selected incidence angles at waiting time $T = 0$ fs.** Insets: Cross sections at selected excitation and detection energies. **a** 2DES map for $\theta = 18°$ displaying diagonal and cross peaks at the LP (1.571 eV) and UP (1.70 eV) energies. In addition to the dispersive "uncoupled" X peak (1.593 eV), cross peaks between both polaritons and the X transition are observed. The deduced resonance energies are marked as dashed lines. **b** The same general features are also seen at $\theta = 21°$. The resonance energies of the polariton peaks are shifted according to their dispersion relation. **c** For $\theta = 27°$,

the detuning between LP and X is sufficiently large to resolve the dispersive line shape of all four diagonal and cross peaks of LP and X. At this angle, the intensities of the UP-related peaks are too weak to be seen. **d** Dispersion relation of one-quantum (1Q) and two-quantum (2Q) excitations, as deduced from 2DES maps (open circles). The 1Q energies match well the linear dispersion relation from Fig. 1e (solid lines). 2Q excitations of lower polaritons (2LP) and "uncoupled" excitons (XX) are extracted together with mixed LP/X and UP/X peaks, while 2UP and UP/LP peaks are lacking.

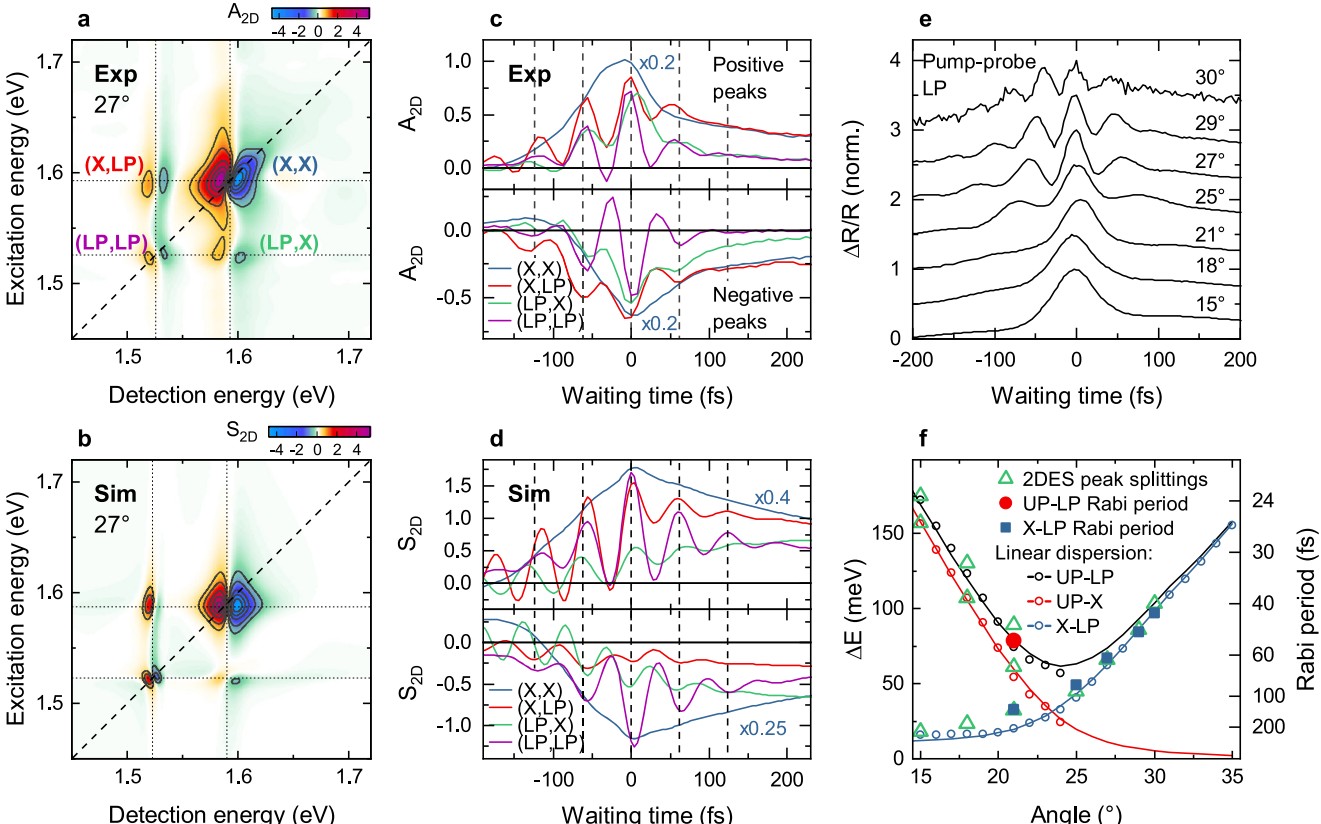

**Fig. 3 | 2DES dynamics revealing polariton Rabi oscillations. a** Experimental 2DES map at $T = 0$ fs and $\theta = 27°$. **b** Simulation of the 2DES map using the effective coupling Hamiltonian introduced in the main text. As in the experiment, diagonal and cross peaks with dispersive line shape involving the LP and X transitions appear, while the UP-related peaks are too weak. **c** The waiting time dynamics of the (LP,LP) diagonal and (LP,X) / (X,LP) cross peak reveal pronounced Rabi oscillations with a period matching the peak splitting while such oscillations are lacking at the (X,X) diagonal. The dynamics detected on the positive and negative sides of the dispersive peaks are displayed in the upper and lower panels, respectively.

**d** Simulations of the waiting time dynamics for all peaks shown in (**c**). **e** Angle-resolved pump-probe dynamics detected at the positive peak of the dispersive LP resonance. **f** Rabi oscillations periods $2\pi\hbar/\Delta E$ (filled symbols) extracted from the 2DES waiting time dynamics for different incidence angles. UP-LP oscillations (red circle) are only resolved at $\theta = 21°$, near the crossing angle. All other periods (blue squares) match the X-LP splitting. The 2DES peak splittings (green triangles) follow the energy differences $\Delta E$ between the corresponding UP, LP and X transitions, as deduced from the linear dispersion in Fig. 1e (open circles and lines).

again reveal $T_X$ oscillations with a period that decreases monotonically with increasing angle. For $\theta \gtrsim 21°$, also the 2DES maps show persistent amplitude oscillations, except for (X,X) (Fig. S4). These oscillations appear predominantly with a period given by the X-LP splitting, as can be seen by comparing the measured oscillation periods (blue squares in Fig. 3f) to those predicted by the linear dispersion (blue line). Only at one selected angle, close to the crossing, we find an oscillation at the anticipated UP-LP splitting (red circle in Fig. 3f and Fig. S19). To explain these observations, we first introduce a phenomenological extension of the TC model that takes the spatial characteristics of our sample into account. For this, we consider two classes of spatially separated J-aggregated excitons. Excitons in the slit region, $X_S$, collectively interact with the plasmon field with a coupling strength $V_S$. In contrast, those excitons, $X_W$, that lie between the slits, on the flat gold film, interact with $V_W$. Both plasmons and excitons are treated as bosonic oscillators[35]. A nonlinearity of the system arises by introducing a finite blue-shift $\Delta E$ of both two-exciton states (Fig. S15). Using this model, 2DES spectra are simulated by solving the Lindblad master equation for the density matrix of the coupled system. As can be seen in Fig. 3b, d, the model quantitatively accounts for our experimental observations. Specifically, the simulations show dispersive peaks with pronounced amplitude oscillations at the period $T_X$, given by the X-LP splitting, while Rabi oscillations at $T_R$ are much weaker in amplitude. As in the experiment, the oscillations are basically absent at (X,X).

Reasonable agreement between experiment and simulation is achieved when choosing $V_S \simeq 3V_W$, with $\hbar\Omega_R = \sqrt{V_S^2 + V_W^2}$, and $\mu_P \simeq \mu_W \simeq 3\mu_S$ (see Section 9 of the Supplementary Information).

To rationalize the microscopic dynamical processes that give rise to these transient 2DES spectra we further invoke an elementary Frenkel exciton model[41,42]. We consider a disordered chain of squaraine molecules, each treated as a fermionic two-level system. Neighboring molecules are dipole-coupled via their optical near fields and interact with the plasmonic mode that is delocalized along the chain. In agreement with FDTD simulations of our sample (Figs. S6–S12), we consider a spatially inhomogeneous plasmon field with a local field in the slit region that is ten times larger than in the region between the slits (Fig. 4a). In the absence of the plasmon field (Fig. S13), the nearest-neighbor coupling results in the formation of few superradiant, moderately localized J-aggregated excitons, strongly red-shifted in energy and extending over ~25 molecules, together with a large number of dark excitons. Between the slits, the wavefunctions of these localized excitons (X) remain basically unchanged in the presence of the coupling to the plasmon mode – except for a minor admixture of excitons inside the slits. The plasmon contribution to their wavefunction is small. In contrast, the superradiant excitons inside the slits couple strongly to the plasmonic mode, resulting in a LP mode carrying substantial contributions from $X_S$ and $P$ and much weaker

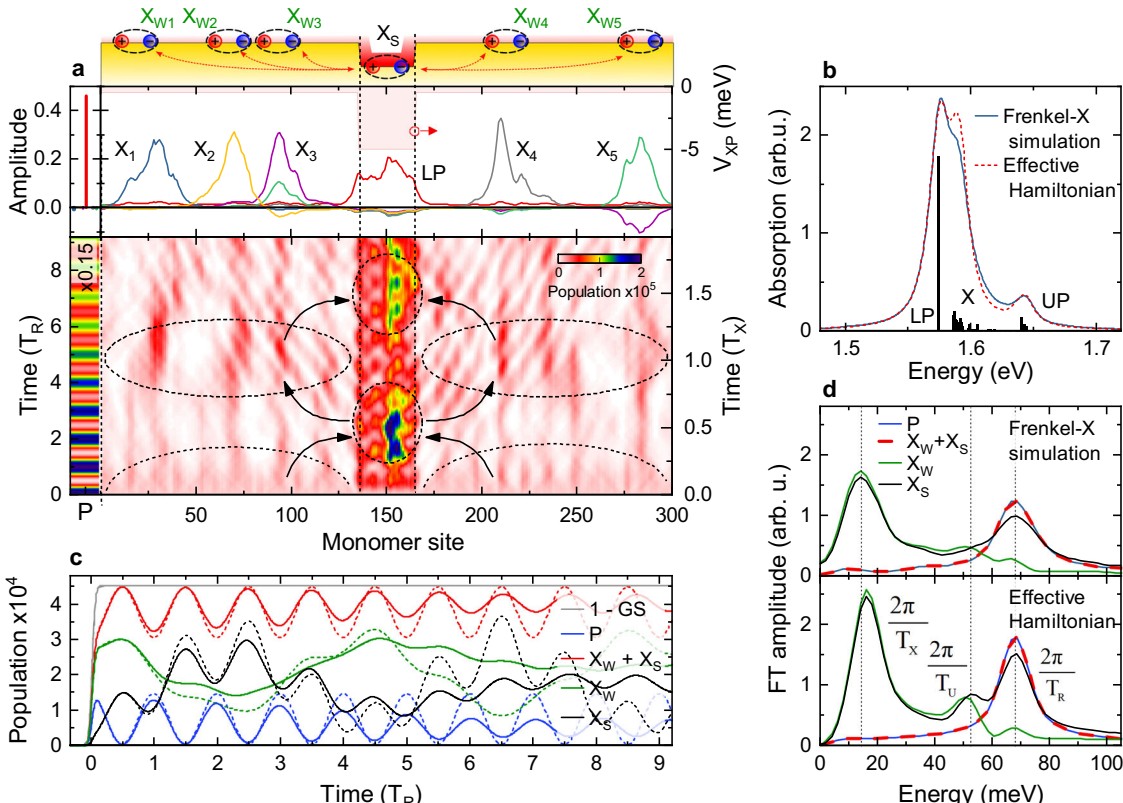

**Fig. 4 | Frenkel exciton simulations of plasmon-driven coherent exciton population oscillations (CPOs). a** A chain of 300 squaraine monomers with disordered site energies and nearest-neighbor coupling forms localized J-aggregate excitons. The monomers are coupled to a delocalized plasmonic mode with a spatially inhomogeneous coupling strength $V_{XP}$ that decreases in amplitude from 5.0 meV for excitons ($X_S$) inside the narrow slit region to 0.5 meV for those ($X_W$) outside the slits. This leads to the absorption spectrum in (**b**) displaying a single, delocalized LP mode and several disordered UP transitions together with an "uncoupled" exciton peak X. Excitons and plasmon mode (P) are resonantly excited by a 5-fs pulse. After excitation, the population of P displays oscillations at a period

$T_R = 2\pi\hbar/(E_{UP} - E_{LP})$, given by the UP-LP splitting. Out-of-phase UP-LP oscillations are seen for the slit excitons. These are superimposed by slower oscillations at $T_X = 2\pi\hbar/(E_X - E_{LP})$. These reflect CPOs from outside the slits into the slit region and back, as illustrated by the arrows (**a**). **c** Population dynamics of the $X_S$, $X_W$ and P states after impulsive excitation, displaying oscillations at $T_R$ and $T_X$. CPOs are seen on both, $X_S$ and $X_W$, but are fully absent for P and $X_S + X_W$. Dynamics for the effective Hamiltonian are shown as dashed lines. **d** Fourier transform of the populations displaying CPOs at $T_X$, Rabi oscillations at $T_R$, and weaker oscillations at $T_U = 2\pi\hbar/(E_{UP} - E_X)$.

contributions from all $X_W$. For LP, all wavefunctions interfere constructively while for the X states, the contributions from $X_S$ and $X_W$ interfere destructively. The resulting linear optical absorption (Fig. 4b) shows strong contributions from the energetically isolated LP state, while the X peak is inhomogeneously broadened. The UP absorption is much weaker since, in our sample, the dipole moment of P ($\mu_P$) and of the sum of all excitons ($\mu_W$ and $\mu_S$) are of similar magnitude. Hence, their emission interferes destructively for the UP peak.

We now discuss the dynamics of the coupled X-SPP system. For this, we impulsively excite all optical resonances with a spatially homogeneous laser field and follow the spatiotemporal evolution of the excited state populations within the chain of squaraine monomers and in the plasmon mode. The plasmon mode shows the expected Rabi oscillations with $T_R$. Out-of-phase oscillations at $T_R$ are most pronounced for the slit excitons $X_S$. They are superimposed, however, with slower oscillations of the $X_S$ population with a period $T_X$. While these slower oscillations are completely absent in the plasmon dynamics, they reappear, phase-shifted by $\pi$, for those excitons, $X_W$, that are localized between the slits. These two distinct types of population oscillations are most clearly seen when spatially integrating over the localized exciton populations $X_S$ and $X_W$ (Fig. 4c). Now, the Rabi oscillations on P are perfectly matched by out-of-phase oscillations of the total population of all excitons, $X_S + X_W$ (red line in Fig. 4c). The plasmon-mediated coherent population oscillations (CPO) between $X_S$ and $X_W$ are only seen in the individual exciton subsystems,

while they are absent in the net exciton population. These model calculations suggest that the dipolar coupling to the plasmon induces spatial oscillations in the exciton density, from outside the slits into the slit region and back. These oscillations appear at the period $T_X$, given by the energy splitting between X and LP. They clearly dominate the exciton dynamics in the region between the slits. Here, the effect of the "traditional" Rabi oscillations with period $T_R$ is weak due to the small local plasmon field amplitude. These conclusions still hold when choosing a more realistic SPP field distribution (Fig. S14) in the simulations. Even when including spatial inhomogeneities both inside and outside the slits, we still observe two distinct classes of Rabi oscillations.

These conclusions are largely corroborated by Fourier transforms of the population dynamics (Fig. 4d). They emphasize the presence of fast Rabi oscillations with $T_R$ and absence of $T_X$ oscillations in the dynamics of the plasmons and of the total exciton population, $X_S + X_W$, (blue and red line), respectively. In contrast, the slower oscillations with $T_X$ between the two distinct classes of excitons become apparent when examining the individual exciton dynamics. These Frenkel exciton simulations form a convincing microscopic basis for the phenomenological extension of the TC model introduced above. Essentially, we can explain the suppression of "traditional" exciton-plasmon Rabi oscillations ($T_R$) and the emergence of CPOs with a longer period $T_X$ in the waiting time dynamics of the 2DES maps by considering two spatially distinct classes of localized J-aggregated

excitons that are mutually coupled to a spatially structured plasmonic mode. This model convincingly accounts for the rich spectral and dynamic features in all angle-dependent 2DES maps. Most importantly, it explains how a spatially delocalized plasmon mode induces a coherent real-space energy transport between spatially separated exciton sites that persists during the coherence time of the strongly coupled system. This maps the complex real space dynamics of a nanostructured system of molecular excitons and plasmons onto an effective three-level system, in which two of the levels, $X_S$, the excitons inside the slits, and $X_W$, the excitons between the slits, are both coupled to a third state, the plasmon. The collective coupling strength of the excitons inside the slits is roughly three times larger than that for those outside the slits. This spatially structured coupling gives rise to CPOs between two of these states without affecting the dynamics of the third. Previously, such CPOs have been discussed in atomic and molecular three-level systems in the context of slow light generation[47] and light storage[48]. Here, we report CPOs in a prototypical all-solid 3-level-system at room temperature and demonstrate how they enable an efficient coherent transport of excitons over mesoscopic distances, from regions outside to inside the slits and back. Atomic and molecular 3-level and 4-level systems offer exciting resources for quantum state manipulation and information processing. The reduction in the speed of light by electromagnetically induced transparency[49], the coherent trapping of population in optically dark states by stimulated Raman adiabatic passage[50] or lasing without inversion[51] are among the manifestations of the control of optical information that can be achieved. We therefore anticipate that the demonstration of related coherent phenomena in all-solid-state systems will open up new avenues towards optical information processing in strongly coupled exciton-plasmon systems. Our results show that strongly coupled exciton-plasmon systems offer exciting new prospects for manipulating coherent quantum transport by light. To leverage these opportunities, direct spatial and temporal visualization of the exciton transport dynamics will be an important next step. Advanced experimental techniques such as ultrafast photoemission electron microscopy, scanning near-field optical microscopy or 2DES microscopy may, in principle, provide the necessary spatio-temporal resolution. Such experiments may give much new insight into coherent quantum transport phenomena on mesoscopic length scales.

## Methods

### Sample preparation

A polycrystalline 200-nm gold film was deposited on a fused silica substrate and subsequently annealed[52,53]. Nanoslit arrays ($80 \times 50 \, \mu m^2$) with a grating period of 530 nm and a slit depth and width of 45 nm were fabricated using focused Ga ion beam milling (Helios NanoLab 600i, FEI). The nanostructured sample was coated with a 10-nm thick J-aggregate thinfilm by spin-coating a solution of squaraine molecules, dissolved in chloroform, following the procedure reported previously[41]. The squaraine molecules, abbreviated as ProSQ-C16, are (S,S)-enantiomers of 2,4-Bis[4-((S))−2-(hexadecyloxymethyl)-pyrroli-done-2,6-dihydroxyphenyl] and have been synthesized as described by Schulz et al.[40,54]. Angle-dependent linear optical characterization of the fabricated samples have been performed using a supercontinuum white light source (SC400-4, Fianium) that was focused onto the structured area under an angle $\theta$ with a polarization perpendicular to the slit orientation (as depicted in Fig. 1a). The reflected light was measured with a fiber spectrometer (FLAME, OceanOptics) and normalized to the reflection from bare gold.

### Experimental 2DES setup

Angle-dependent pump-probe and 2DES data were recorded with a home-built high-repetition rate setup which allows for rapid data acquisition, resulting in a high signal-to-noise ratio of the measured data within short measurement times. To this aim we employ a high-repetition rate laser system (Tangerine V2, Amplitude Systèms), delivering 260-fs pulses (full width at half maximum of the pulse intensity), centered around 1030 nm, at a repetition rate of 175 kHz. A fraction of the 60 μJ pulse energy is used to pump a home-built non-collinear optical parametric amplifier (NOPA), previously described in Ref. 41 and based on a design published in Ref. 55. The NOPA outputs 650−900 nm pulses with a measured pulse duration of ~12 fs, as characterized using second harmonic frequency resolved optical gating (Fig. S2). The NOPA pulses are used in a home-built 2DES setup, also previously reported in Ref. 41. A phase-stable and collinear pump-pulse pair with variable delay $\tau$ (coherence time) is generated by an interferometer based on birefringent wedges (Translating Wedge-based Identical pulse eNcoding System, TWINS[56]). The pump pulses are periodically switched on and off at 43.75 kHz by a mechanical chopper system (MC2000B, Thorlabs) equipped with a custom-made blade (500 slots). The vertically aligned and p-polarized pump and probe beams are focused onto the sample to a spot size of ~$60 \times 60 \, \mu m^2$ under the same angle of incidence $\theta$ and with a small angle mismatch in the orthogonal direction (see Fig. 1a). The sample is mounted on a rotation stage and the axis of rotation is tuned to coincide with the location of the nanostructured sample to allow for conveniently and accurately tuning $\theta$ for both pump and probe in the angle-dependent 2DES and pump-probe experiments. The reflected probe beam is then collected and sent to a monochromator (Acton SP2150i, Princeton Instruments) with an attached fast line camera (Aviiva EM4, e2v) allowing to rapidly record probe spectra S at an acquisition rate of 87.5 kHz (i.e., at half the laser repetition rate) as a function of the detection energy $E_{det}$. We thus record a differential reflectivity spectrum $\Delta R/R$ from a set of 4 laser shots by taking the difference between spectra with ($S_{on}$) and without ($S_{off}$) pump

$$\frac{\Delta R}{R}(\tau, T, E_{det}) = \frac{S_{on}(\tau, T, E_{det}) - S_{off}(E_{det})}{S_{off}(E_{det})} \quad (1)$$

Here, the waiting time $T$ denotes the delay between the second pump and the probe pulse. This delay is controlled using a motorized linear translation stage (M126.DG1, Physik Instrumente). For the 2DES measurements, at each waiting time $T$, the coherence time is scanned and differential spectra are recorded on the fly, whereas for the pump-probe experiments, we fix the coherence time to $\tau = 0$ fs and scan only the waiting time $T$.

To calculate absorptive 2DES maps[57]

$$A_{2D}(E_{ex}, T, E_{det}) = \Re\left(\int_{-\infty}^{\infty} \Theta(\tau) \frac{\Delta R}{R}(\tau, T, E_{det}) e^{iE_{ex}\tau/\hbar} d\tau\right) \quad (2)$$

a Fourier transform of the differential reflectivity is performed along the coherence time to obtain the 2DES spectra as a function of the excitation energy axis $E_{ex}$ ($\hbar$ Planck's reduced constant, $\Theta(\tau)$ Heaviside step function).

### Frenkel exciton simulations

We model the coupling of the molecular J-aggregate with the plasmon mode by performing qualitative microscopic simulations based on a disordered Frenkel exciton model. We choose $N = 300$ squaraine monomer states at $E_{SQ} = 1.923$ eV with Gaussian disorder in site energy of $\sigma = 15.4$ meV[41]. We set the monomer transition dipole moment to $\mu_{SQ} = 1/\sqrt{N}$.

The plasmon mode is introduced as a single state $|P\rangle$ with energy $E_P$ and transition dipole moment $\mu_P$. The Frenkel exciton

Hamiltonian reads

$$\hat{H}_F = E_P |P\rangle\langle P| + \sum_{n=1}^{N}\left(E_n |n\rangle\langle n| + V_{XP,n}(|n\rangle\langle P| + |P\rangle\langle n|)\right) + \sum_{n,m=1}^{N} J_{nm}|n\rangle\langle m| \tag{3}$$

where $V_{XP,n} = -\boldsymbol{\mu_{SQ}} \cdot \mathbf{E_{SPP,n}}$ denotes a local exciton plasmon coupling between the monomer state $|n\rangle$ at site n and the plasmon field $\mathbf{E_{SPP,n}}$ at the same site. We choose a dipolar coupling between monomer states of $J = -166$ meV, and limit this coupling to nearest neighbors by setting $J_{n,m} = J\delta_{n,m\pm1}$. Periodic boundary conditions are applied. For $V_{XP,n} = 0$, Eq. (3) thus describes a superradiant J-aggregate exciton state with $E_X = 1.59$ eV determined by the nearest-neighbor coupling between the monomers (Fig. S13b)[41,42].

We choose the local X-P couplings $V_{XP,n}$ to be proportional to the amplitude of the plasmon field at each monomer site n. We assume that the local coupling is governed by the in-plane components of the local SPP field. To account for the spatial profile of the SPP field the X-P coupling is divided into two regions: $V_{S,n}$ for the strong coupling inside the slits and $V_{W,n}$ for a weaker coupling in the region in-between two slits. $V_{S,n}$ and $V_{W,n}$ are taken as constant and real-valued parameters. The number of monomers in the two regions are $N_S$ and $N_W$, respectively.

For rationalizing the results of this Frenkel exciton model, we introduce an effective Hamiltonian

$$\hat{H}_{red} = E_P |P\rangle\langle P| + E_{X_S}|X_S\rangle\langle X_S| + E_{X_W}|X_W\rangle\langle X_W| \\ + V_S(|X_S\rangle\langle P| + |P\rangle\langle X_S|) + V_W(|X_W\rangle\langle P| + |P\rangle\langle X_W|) \tag{4}$$

comprising four states $|0\rangle, |P\rangle, |X_S\rangle$ and $|X_W\rangle$, where $|X_S\rangle$ and $|X_W\rangle$ indicate strongly coupled (inside the slits) and weakly coupled excitons (in between two slits), respectively. The ground state energy of the system is set to zero. The exciton energies are approximated as $E_{X_S} = E_{X_W} = E_{SQ} - 2J$. The collective transition dipole moments and coupling strengths can be calculated as $\mu_{X_S} = \sqrt{N_S}\mu_{SQ}$ and $\mu_{X_W} = \sqrt{N_W}\mu_{SQ}$ for the transition dipole moments, and $V_S = \sqrt{N_S}V_{S,n}$ and $V_W = \sqrt{N_W}V_{W,n}$ for the coupling strengths.

We simulate the dynamics following optical excitation by numerically integrating the master equation in Lindblad form[57,58]

$$\dot{\hat{\rho}} = -\frac{i}{\hbar}\left[\hat{H},\hat{\rho}\right] + \frac{1}{2}\sum_k \left(2\hat{L}_k\hat{\rho}\hat{L}_k^\dagger - \hat{L}_k^\dagger\hat{L}_k\hat{\rho} - \hat{\rho}\hat{L}_k^\dagger\hat{L}_k\right) \tag{5}$$

$\hat{H} = \hat{H}_S + \hat{H}_{\text{int}}(t)$ describes the free evolution of the system via $\hat{H}_S$ and its light-matter interaction is governed by the time-dependent interaction Hamiltonian

$$\hat{H}_{\text{int}}(t) = -\hat{\mu}E(t) \tag{6}$$

which accounts for optical excitation of the system by an external light field in dipole approximation. $\hat{\mu}$ denotes the transition dipole moment operator of the system. We assume a short and sufficiently weak 5-fs pulse at 1.6 eV. Dephasing and relaxation processes are incorporated through appropriate Lindblad operators $\hat{L}_k$[58,59]. For more details see Supplementary Information section 8.

**Simulation of nonlinear signals**
To simulate the experimental pump-probe and 2DES maps, we compute the full dynamics of the density matrix $\hat{\rho}$ considering a series of interactions with up to three laser pulses. This allows us to calculate the sample polarization $P(t) = \text{Tr}(\hat{\mu}\hat{\rho}(t))$ and the resulting optical spectra[39].

For the numerical integration of the master equation in Lindblad form, Eq. (5), we use a non-perturbative approach[58,59]. The total electric

field, entering the interaction Hamiltonian in Eq. (6),

$$E(t) = \sum_{\substack{n=pu1, \\ pu2,pr}} E_{0,n}\,e^{-2\ln2\left(\frac{t-t_n'}{\Delta t}\right)^2}\cos\left(\omega_L\left(t-t_n'\right)+\phi_n\right) \tag{7}$$

comprises of up to three laser pulses (pump 1, pump 2 and probe) with amplitude $E_{0,n}$, pulse duration $\Delta t = 10$ fs, frequency $\omega_L = 1.6$ eV$/\hbar$, phase $\phi_n$ for phase cycling and a time shift $t_n'$. We label the delay between the pump pulses as the coherence time $\tau = t_{pu2}' - t_{pu1}'$ and the delay between the second pump pulse and the probe pulse as the waiting time $T = t_{pr}' - t_{pu2}'$. A detection time t of zero corresponds to the arrival time of the probe pulse. The detection time axis coincides with the one used for the numerical integration of Eq. (5).

In the simulations, we calculate the linear polarization P$^{(1)}$(t) without pump, or a total polarization $P^{tot}(\tau,T,t)$, including all nonlinear signals that arise from all possible interactions with the three pulses[39]— we then deduce linear and total susceptibilities as a function of the detection energy $E_{det}$ from these polarizations via a Fourier transform along the detection time t

$$\chi^{(1)}(E_{det}) = \frac{1}{\varepsilon_0}\mathscr{F}\left(P^{(1)}(t)\right)/\mathscr{F}\left(E_{pr}(t)\right) \tag{8}$$

$$\chi^{tot}(\tau,T,E_{det}) = \frac{1}{\varepsilon_0}\mathscr{F}\left(P^{tot}(\tau,T,t)\right)/\mathscr{F}\left(E_{pr}(t)\right) \tag{9}$$

where $\varepsilon_0$ denotes the vacuum dielectric constant.

We employ a phase-cycling scheme in our simulations to isolate the desired linear and nonlinear contributions to our signal. For this, we calculate $P^{tot}(\tau,T,t)$ for four different phase settings of $\phi_{pu1} = \phi_{pu2} = \left[0,\frac{\pi}{2},\pi,\frac{3\pi}{2}\right]$ for both pump pulses, with $\phi_{pr} = 0$[60–62]. We then perform the average over these four settings to obtain the phase-cycled average $\chi_{PC}^{tot}(\tau,T,E_{det})$. This allows for extracting the contributions to the nonlinear signal corresponding to those that are measured in the partially collinear experimental geometry. We then obtain the nonlinear signal as

$$\chi^{nl}(\tau,T,E_{det}) = \chi_{PC}^{tot}(\tau,T,E_{det}) - \chi^{(1)}(E_{det}) \tag{10}$$

From this nonlinear susceptibility, we calculate the 2DES map as a function of the coherence time as

$$S_{2D}(\tau,T,E_{det}) = \Im\left(\chi^{nl}(\tau,T,E_{det})\right) \tag{11}$$

Finally, energy-energy 2DES maps for a fixed waiting time T are obtained by taking the real part of the Fourier transform along the coherence time τ

$$S_{2D}(E_{ex},T,E_{det}) = \Re\left(\int_{-\infty}^{\infty}\theta(\tau)S_{2D}(\tau,T,E_{det})e^{iE_{ex}\tau/\hbar}d\tau\right) \tag{12}$$

yielding the excitation energy axis $E_{ex}$. Setting $\tau = 0$ in the simulations allows for calculating pump-probe spectra as a function of $E_{det}$ and $T$.

The exciton-plasmon system is modeled based on the previously introduced effective Hamiltonian and contains, in addition to the ground state, not only the three one-quantum states $|P\rangle, |X_S\rangle$ and $|X_W\rangle$, but also six two-quantum states $|2P\rangle, |P,X_S\rangle, |P,X_W\rangle, |XX_S\rangle, |X_S,X_W\rangle$ and $|XX_W\rangle$. The plasmon and exciton subsystems are each treated as a harmonic oscillator. To account for the exciton non-linearity, the two-exciton state is blue-shifted by $\Delta E = 5$ meV[41,42]. Dipolar interactions of the strongly ($V_S$) and weakly ($V_W$) coupled excitons with the plasmon field are introduced by adding two coupling

Hamiltonians in rotating wave approximation, following[35]

$$\hat{H}_{X_W P} = V_W \left( \hat{b}_P^\dagger \hat{b}_{X_W} + \hat{b}_{X_W}^\dagger \hat{b}_P \right) \tag{13}$$

$$\hat{H}_{X_S P} = V_S \left( \hat{b}_P^\dagger \hat{b}_{X_S} + \hat{b}_{X_S}^\dagger \hat{b}_P \right) \tag{14}$$

with $\hat{b}^\dagger$ and $\hat{b}$ being the creation and annihilation operators. Radiative damping and pure dephasing phenomena are included using the Lindblad formalism. See Supplementary Information section 9 for more details.

## Data availability
All experimental and simulation data supporting the findings are presented in the paper and Supplementary Information in graphic form. Source data will be provided by the corresponding authors upon request.

## Code availability
All codes are described in the paper or Supplementary Information and will be provided by the corresponding authors upon request.

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

## Acknowledgements

We acknowledge the financial support from Deutsche Forschungsgemeinschaft within the Priority Program Tailored Disorder (SPP1839) and SFB1372/2-Sig01, INST 184/163-1, INST 184/164-1, Li 580/16-1, and DE 3578/3-1, as well as the Graduate Schools "Molecular Basis of Sensory Biology" (RTG 1885) and "Template-designed organic electronics (TIDE)" (GRK 2591). Financial support from the Niedersächsische Ministerium für Wissenschaft und Kultur ("DyNano") and the Volkswagen Foundation (SMART) is also gratefully acknowledged. M.S., M.G. and S.S. wish to thank the BMBF for support (NanoMatFutur FKZ: 13 N13637). M.S. and S.S. further acknowledge financial support from the BMBF within the project tubLAN Q.0. J.-H.Z. acknowledges a Carl von Ossietzky Fellowship from University of Oldenburg and thanks the Alexander von Humboldt Postdoctoral Fellowship for personal support. S.T. acknowledges support of the Humboldt Research Award (Germany). Y.Z. acknowledges the support from the Laboratory Directed Research and Development (LDRD) program of Los Alamos National Laboratory. S.T. and Y.Z. acknowledge the support from US DOE, Office of Science, Basic Energy Sciences, Chemical Sciences, Geosciences, and Biosciences Division under Triad National Security, LLC ("Triad") contract Grant 89233218CNA000001 (FWP: LANLE3F2). LANL is operated by Triad National Security, LLC, for the National Nuclear Security Administration of the U.S. Department of Energy (contract no. 89233218CNA000001). M.F.S. thanks the Manchot Foundation for a doctoral scholarship.

## Author contributions

D.T., M.G., T.Q. and S.S. designed and M.G. fabricated the nanostructured sample. M.F.S. and A.L. synthesized the squaraine molecules. D.T., M.G., T.Q. and J.-H.Z. constructed the experimental setup and performed the experiments. D.T., M.G. and T.Q. analyzed the experimental results. S.S. and M.S. performed the FDTD simulations. D.T. and M.G. performed the simulations of nonlinear signals. D.T., M.G., Y.Z. and S.T. realized the Frenkel-exciton simulations. C.L., D.T. and M.G. conceived the project. C.L. and A.D.S. supervised the project. C.L., A.D.S., D.T. and M.G. wrote the manuscript. All authors contributed to discussions and gave input to the manuscript writing.

## Funding

## Competing interests

The authors declare no competing interests.
