## [Peer Review File · Nature Communications]

Plasmon mediated coherent population oscillations in molecular aggregatesREVIEWER COMMENTS

Reviewer #1 (Remarks to the Author):

In this manuscript Timmer and co-workers present a very interesting study, mainly experimental, of the dynamics of plasmon-exciton polaritons, i.e., plexcitons, emerging when the surface plasmons supported by a gold nanoslit array strongly interact with the localized excitons present in organic J-aggregated thin films. For that, they use a novel ultrafast two-dimensional electronic spectroscopy technique that allows them to analyse in depth the temporal response and evolution of the hybrid light-matter states as a function of energy thanks to the pump-probe character of the experimental technique. The experimental results are fascinating as they provide very relevant information on the nature of organic polaritons. The research area of organic polaritonics has bloomed in recent years as it is thought that polaritonic systems could be utilized in opto-electronic devices and could also find feasible applications in the emergent field of quantum-based technologies.

The manuscript is very well-written, and it is easy to follow even for a non-specialized reader. The results are relevant and timely in the research for exploring the exciting properties of organic polaritons and, therefore, I support the publication of this manuscript in Nature Communications. My only concern regarding the work presented in this manuscript is related to the theoretical model they develop to understand the results presented in Figure 3, namely, the presence of polariton Rabi oscillations as a function of waiting time. The model assumes that there are just two classes of spatially separated J-aggregated excitons: i) excitons in the slit regions, and ii) excitons located between the slits. Intuitively, one would expect that instead of just two classes of excitons, there would be a continuum of excitons distributed along the unit cell of the nanoslit array, accompanying the electric field profile of the surface plasmons. I recommend the authors to comment in the new version of the manuscript how their conclusions regarding the emergence of a plasmon-driven exciton population transfer could change when a more realistic model is applied.

Reviewer #2 (Remarks to the Author):

The manuscript by D. Timmer et al. has been carefully revised. In this paper the authors perform two dimensional electronic spectroscopy measurements on a hybrid plasmonic cavity (i.e. a gold nanoslit array covered by a J-aggregated thin film) to study the coupling in real time between J-aggregate excitons and surface plasmon polaritons. The 2DES maps are measured as a function of the incident angle, and show diagonal and cross peaks oscillating in time with a period that match the frequency difference between the exciton and the lower polariton peak for different range of angles. These Rabi oscillations are related to strong coupling between excitons and surface plasmon polaritons. To my knowledge, this is the first time that 2DES has been applied to study coherent coupling in metal-based hybrid systems in the strong coupling regime. In general, the paper is clearly written, the results of experiments are convincing and presented in an understandable manner. The Supplemental Material also provides some valuable information regarding the details of the research. For all these reason, the paper deserves to be published in this journal. However I have few general comments that I would like the authors address.

Below the comments:

1-In one of their previous work, published some years ago, by some of the authors of this manuscript (P. Vasa et al. Nature Photonics 7, 128–132 (2013)), high temporal resolution pump-probe optical measurements have been performed on similar samples (J-aggregate/metal nanostructure) and Rabi oscillations between excitons and SPPs have been measured. The authors should better stress the novelties contained in the manuscript compared with their previous work. What is advantage of using 2DES compared to standard optical pump-probe spectroscopy in these systems?

2-UP-LP oscillations are detectable only at one specific angle (i.e. close to the crossing). How is it determined the red dot reported in figure 3f? Is it just determined from the expected energy difference between UP and LP at that specific angle? or is it extracted from the Fourier transform of the temporal trace in fig. S18? Can the authors show the FT of the trace in fig.S18?

Reviewer #3 (Remarks to the Author):

In this research manuscript, the authors employed a technique known as Two-Dimensional Electronic Spectroscopy (2DES) to investigate the ultrafast coherent dynamics of plasmon-exciton polaritons. Their findings challenged the conventional understanding of these dynamics, as they observed that the peak behaviors could not be solely attributed to the commonly accepted concept of Rabi oscillations, which traditionally describe the energy exchange between plasmons and excitons. Instead, the authors proposed an intriguing alternative explanation. They posited that there is an energy exchange occurring not only between plasmons and excitons but also among excitons situated in distinct surface locations, each exhibiting different coupling strengths.

Importantly, their conclusions were substantiated through clear model simulations. This observation of subtle coherent dynamics represents a notable contribution to the field. It underscores the significance of visualizing and comprehending the intricate dynamics within strongly coupled systems. Moreover, it underscores the limitation of simplistically treating such systems as composed solely of plasmonic and excitonic components at the microscopic level. This novel insight is likely to push further investigations in this area of research. Hence, I propose that this manuscript should be considered for publication in Nature Communications once the aforementioned questions have been more comprehensively addressed.

One particularly surprising aspect of this study was the oscillatory behavior observed between Lower Polariation (LP)band and "uncoupled excitons." Figure 4 suggest that there is an energy transfer transpiring between two distinct types of excitons – those strongly coupled and those weakly coupled – facilitated by the Surface Plasmon Polariton (SPP). An intriguing question arises from this finding: Can this phenomenon of energy transfer be directly observed experimentally? If so, what is the pertinent length scale that governs this transport behavior?

The manuscript mentions that the excitons have a decay time on the order of 100 fs. This leads to another question: Is it the lifetime of the excitons or the plasmons that imposes the ultimate limitation on the coherence of the polaritons?

The model presented in equ.4 indicates that the two types of excitons are independently coupled to the plasmon, each with varying degrees of strength. However, an interesting hypothesis arises from this observation: could it be possible that the excitons are directly coupling to one another, considering that they all reside within the region influenced by the SPP field?

Response to the Reviewers' Reports

We wish to thank the Reviewers for their careful review of our paper and their constructive and helpful comments and recommendations. We are happy to hear the positive feedback of all three Reviewers, supporting publication in Nature Communications.

In the following we answer each of the Reviewers' comments one by one and list the changes we made to the manuscript. We mark the quotes of the Reviewer reports by blue text color and italic font.

Response to Reviewer #1

In this manuscript Timmer and co-workers present a very interesting study, mainly experimental, of the dynamics of plasmon-exciton polaritons, i.e., plexcitons, emerging when the surface plasmons supported by a gold nanoslit array strongly interact with the localized excitons present in organic J-aggregated thin films. For that, they use a novel ultrafast two-dimensional electronic spectroscopy technique that allows them to analyse in depth the temporal response and evolution of the hybrid light-matter states as a function of energy thanks to the pump-probe character of the experimental technique. The experimental results are fascinating as they provide very relevant information on the nature of organic polaritons. The research area of organic polaritonics has bloomed in recent years as it is thought that polaritonic systems could be utilized in opto-electronic devices and could also find feasible applications in the emergent field of quantum-based technologies.

The manuscript is very well-written, and it is easy to follow even for a non-specialized reader. The results are relevant and timely in the research for exploring the exciting properties of organic polaritons and, therefore, I support the publication of this manuscript in Nature Communications.

Reply: We thank the Reviewer for the positive assessment of our research results and for supporting the publication of our manuscript in Nature Communications.

My only concern regarding the work presented in this manuscript is related to the theoretical model they develop to understand the results presented in Figure 3, namely, the presence of polariton Rabi oscillations as a function of waiting time. The model assumes that there are just two classes of spatially separated J-aggregated excitons: i) excitons in the slit regions, and ii) excitons located between the slits. Intuitively, one would expect that instead of just two classes of excitons, there would be a continuum of excitons distributed along the unit cell of the nanoslit array, accompanying the electric field profile of the surface plasmons. I recommend the authors to comment in the new version of the manuscript how their conclusions regarding the emergence of a plasmon-driven exciton population transfer could change when a more realistic model is applied.

Reply: The Reviewer is correct that the model for the simulations of the 2DES dynamics in Fig. 3 indeed assumes only two classes of spatially separated excitons (strongly and weakly coupled). This is of course an oversimplification. In fact, we believe that it is a rather striking result of our experiments that many of the features in Fig. 3 can indeed be explained by such a comparatively simple model.

To convince ourselves, that this simplification is reasonably well justified, we have performed rather extensive Frenkel exciton simulations of the quantum dynamics. These allow us to test in more detail the effects of different spatial distributions of the optical near-field on the exciton dynamics. In Fig. R1 we show simulations considering a spatially **inhomogeneous** field distribution (Fig. R1a, upper panel) that resembles more closely the SPP field obtained by our FDTD simulations (Fig. S10). Qualitatively, the results are very similar to those in Fig. 4. Again, we see two distinct classes of Rabi oscillations: (i) fast oscillations, mainly inside the slits, at the Rabi frequency that corresponds to the polariton splitting in the sample and (ii) a

slower oscillatory exciton transport between regions outside and inside the slits. Fourier transforms clearly show these two distinct classes.

In general, we observe in these simulations that the splitting in “fast” and “slow” Rabi oscillations is robust against variations in the spatial field profile as long as the difference between local field enhancement in the two sample regions is sufficiently large. We think that this can be understood rather intuitively. The strong localized fields in the slits give rise to the usual polariton formation. The weak fields outside the slits have three main effects: (i) they define the amplitude, but not the energy position of the “uncoupled exciton peak” in the linear spectra, (ii) their amplitude affects the polariton splitting in the sample and (iii) it gives rise to the slow Rabi oscillations between the two classes of excitons uncovered in our experiments. This general picture is rather insensitive to the details of the spatial field distribution in the region outside the slits, justifying the simplified exciton model used to simulate the data in Fig. 3. Of course, this simplified picture breaks down in case that the spatial fluctuations in coupling strengths become too large. Obviously, spatially resolved studies of these Rabi oscillations dynamics would be highly desirable to proof these conclusions.

Figure R1: Frenkel exciton simulations of the local exciton density considering a spatially inhomogeneous field distribution in the region outside the slits. This distribution mimics more closely the SPP field distribution in Fig. S10. **a:** Again, we see fast Rabi oscillations in the slits, superimposed by slow oscillations between regions outside and inside the slits. **b:** The Fourier transforms of the population oscillations clearly show the same two distinct classes of Rabi oscillations as in Fig. 4.

Changes: We now add Fig. R1 to Section 8 of the Supporting Information. When discussing the population dynamics of the Frenkel exciton simulations on p. 11 of our manuscript, we now add: “These conclusions still hold when choosing a more realistic SPP field distribution, (Fig. S14) in the simulations. Even when including spatial inhomogeneities both inside and outside the slits, we still observe two distinct classes of Rabi oscillations.”

Response to Reviewer #2

The manuscript by D. Timmer et al. has been carefully revised. In this paper the authors perform two dimensional electronic spectroscopy measurements on a hybrid plasmonic cavity (i.e. a gold nanoslit array covered by a J-aggregated thin film) to study the coupling in real time between J-aggregate excitons and surface plasmon polaritons. The 2DES maps are measured as a function of the incident angle, and show diagonal and cross peaks oscillating in time with a period that match the frequency difference between the exciton and the lower polariton peak for different range of angles. These Rabi oscillations are related to strong coupling between excitons and surface plasmon polaritons. To my knowledge, this is the first time that 2DES has been applied to study coherent coupling in metal-based hybrid systems in the strong coupling regime. In general, the paper is clearly written, the results of experiments are convincing and presented in an understandable manner. The Supplemental Material also provides some valuable information regarding the details of the research. For all these reason, the paper deserves to be published in this journal. However I have few general comments that I would like the authors address.

Reply: We thank the Reviewer for finding our experimental results convincing and the manuscript clearly written. In the following we will address the two comments made by the Reviewer individually.

Below the comments:

1-In one of their previous work, published some years ago, by some of the authors of this manuscript (P. Vasa et al. Nature Photonics 7, 128–132 (2013)), high temporal resolution pump-probe optical measurements have been performed on similar samples (J-aggregate/metal nanostructure) and Rabi oscillations between excitons and SPPs have been measured. The authors should better stress the novelties contained in the manuscript compared with their previous work. What is advantage of using 2DES compared to standard optical pump-probe spectroscopy in these systems?

Reply: Indeed, some of us have published a first time-resolved study of exciton-plasmon Rabi oscillations in the cited paper. In this work, we used pump-probe spectroscopy to trace the oscillations. This implies an impulsive excitation of all resonances by the broadband pump laser. We observed clear signatures of ultrafast Rabi oscillations in the experimental data. The decoherence times in the sample were so short, however, that it was difficult to obtain detailed insight into the microscopic origin of these oscillations from such one-dimensional spectra (Fig. 3 of Vasa et al.). Importantly, it was difficult to separate the role of upper polaritons and uncoupled excitons in these oscillations.

2DES spectra, as reported here, provide access to the quantum dynamics of each resonance that is excited in the system. By adding angle-tuning to 2DES, we can uncover (i) the cross peaks between upper and lower polaritons and (ii) between “uncoupled excitons” and both types of polaritons, for the first time. The oscillatory waiting time dynamics of the 2DES peaks provides the crucial new information about coherent couplings that allowed us to distinguish between the two types of Rabi oscillations.

Changes: On page 8 we have now added: “We will argue below that the oscillatory waiting time dynamics of the 2DES peaks provides crucial new information about coherent couplings in exciton-plasmon systems that has not been obtained from one-dimensional pump-probe spectroscopy, as previously reported by some of the present authors [Vasa et al. Nature Photonics 7, 128–132 (2013)].”

2-UP-LP oscillations are detectable only at one specific angle (i.e. close to the crossing). How is it determined the red dot reported in figure 3f? Is it just determined from the expected energy difference between UP and LP at that specific angle? or is it extracted from the Fourier transform of the temporal trace in fig. S18? Can the authors show the FT of the trace in fig.S18?

Reply: We have obtained the red data point in Fig. 3f from the oscillations in the temporal trace in Fig. S18 (S19 in the revised version), as indicated by vertical dashed lines. The oscillation period of 53 fs corresponds to a normal mode splitting of 78 meV. The same oscillation period of 53 fs (78 meV) is obtained from a peak analysis of the Fourier transform of this trace, as now shown in Fig. R2b.

Changes: We now add the Fourier transform shown in Fig. R2b to Fig. S19.

Figure R2: Observation of UP-LP Rabi oscillations at $\theta = 21^\circ$. **a:** Dynamics of the UP-LP cross-peak in the 2DES measurement. Vertical lines mark oscillations on the temporal trace with 53 fs period, corresponding to an energetic splitting between UP and LP of 78 meV. **b:** Fourier transform of the residuals in panel a after subtracting the slow background (red dotted line). The dashed line marks the peak energy of 78 meV that corresponds to the UP-LP Rabi oscillation period of 53 fs.

Response to Reviewer #3

In this research manuscript, the authors employed a technique known as Two-Dimensional Electronic Spectroscopy (2DES) to investigate the ultrafast coherent dynamics of plasmon-exciton polaritons. Their findings challenged the conventional understanding of these dynamics, as they observed that the peak behaviors could not be solely attributed to the commonly accepted concept of Rabi oscillations, which traditionally describe the energy exchange between plasmons and excitons. Instead, the authors proposed an intriguing alternative explanation. They posited that there is an energy exchange occurring not only between plasmons and excitons but also among excitons situated in distinct surface locations, each exhibiting different coupling strengths.

Importantly, their conclusions were substantiated through clear model simulations. This observation of subtle coherent dynamics represents a notable contribution to the field. It underscores the significance of visualizing and comprehending the intricate dynamics within strongly coupled systems. Moreover, it underscores the limitation of simplistically treating such systems as composed solely of plasmonic and excitonic components at the microscopic level. This novel insight is likely to push further investigations in this area of research. Hence, I propose that this manuscript should be considered for publication in Nature Communications once the aforementioned questions have been more comprehensively addressed.

Reply: We thank the Reviewer for the positive feedback. In the following we will address the questions of the Reviewer individually.

One particularly surprising aspect of this study was the oscillatory behavior observed between Lower Polariton (LP) band and "uncoupled excitons." Figure 4 suggest that there is an energy transfer transpiring between two distinct types of excitons – those strongly coupled and those weakly coupled – facilitated by the Surface Plasmon Polariton (SPP). An intriguing question arises from this finding: Can this phenomenon of energy transfer be directly observed experimentally? If so, what is the pertinent length scale that governs this transport behavior?

Reply: This is an excellent point. In our sample, the pertinent length scale of this energy transfer is given by the average spatial separation of strongly and weakly coupled excitons. This is half the grating period of 530 nm of our sample. In general, it is given by the coherence lengths of the different polariton excitations in the sample. Not only upper and lower polaritons arise from a coupling of localized excitons of the aggregate to a delocalized plasmon field. The observation of coherent population oscillations implies that also the “weakly coupled excitons” are delocalized across the unit cell due to their coupling to the plasmons. The Frenkel exciton simulations suggest that this coherent energy transfer can be observed directly by probing the exciton dynamics in experiments that provide a combined high time resolution of ~ 10 fs and a spatial resolution of a few tens of nm. Such experiments are challenging but could, in principle, be performed by using either ultrafast photoelectron emission microscopy or scanning near-field optical

microscopy. Experiments in this direction are currently underway in our laboratory. Ultrafast 2DES far-field microscopy may, in principle, also be an option. The spatial resolution that has been demonstrated so far, e.g., in experiments in the Cundiff group, may not yet be sufficient to resolve the coherent transport dynamics.

Changes: At the end of our manuscript, we now state: “Our results show that strongly coupled exciton-plasmon systems offer exciting new prospects for manipulating coherent quantum transport by light. To leverage these opportunities, direct spatial and temporal visualization of the exciton transport dynamics will be an important next step. Advanced experimental techniques such as ultrafast photoemission electron microscopy, scanning near-field optical microscopy or 2DES microscopy may, in principle, provide the necessary spatio-temporal resolution. Such experiments may give much new insight into coherent quantum transport phenomena on mesoscopic length scales.”

The manuscript mentions that the excitons have a decay time on the order of 100 fs. This leads to another question: Is it the lifetime of the excitons or the plasmons that imposes the ultimate limitation on the coherence of the polaritons?

Reply: This is a highly interesting, yet highly complex question. Both excitons and plasmons couple strongly to light and, thus, radiative damping processes ultimately limit their lifetimes. In our aggregated samples, exciton coherence is limited by pure dephasing due to fluctuations of the excitonic dipole moments. In principle, however, these can be suppressed at low temperatures. For the plasmons, Ohmic losses certainly limit the lifetime. These losses can be reduced, however, by tailoring the mode profile of the plasmonic mode and limiting the fraction of the mode inside the metal (see C. Ropers et al. PRL 94, 143901 showing plasmon lifetimes of > 200 fs in related systems). Hence, the ultimate limitation may indeed come from radiative damping phenomena. As such, one may expect, that in a strongly coupled system, cooperative damping phenomena, i.e. sub- and superradiance, impose the ultimate limitation on the coherence of the polaritons. So far, however, there are very few experimental studies of such effects that are known to us (some steps of our own in this direction are published in W. Wang et al., ACS Nano 8, 1056 (2014)).

Our experimental results show no clear signs of such cooperative damping effects. For the investigated samples, we estimate dephasing time of the excitons of ~ 80 fs, while their radiative times are much longer. The lifetimes of the SPP modes are also in the range of 100 fs. The polariton seem to have similar dephasing times. Therefore, we did not explicitly discuss polariton dephasing in the main manuscript. The damping properties assumed in the density matrix simulations are given in the Supporting Information. As such, the point that is raised by the Reviewer is very interesting but goes far beyond the discussion in our manuscript. We honestly think that it is not so easy to cover the subtle aspects of this discussion in a short paragraph in the manuscript without disrupting its flow. Therefore, we suggest to keep the manuscript as it is now. We will try to provide a more in-depth discussion once experimental data providing more detailed insights are available.

The model presented in equ.4 indicates that the two types of excitons are independently coupled to the plasmon, each with varying degrees of strength. However, an interesting hypothesis arises from this observation: could it be possible that the excitons are directly coupling to one another, considering that they all reside within the region influenced by the SPP field?

Reply: The reduced Hamiltonian in Eq. 4 indeed treats the two classes of excitons as independent and non-interacting. Their interaction occurs indirectly and via the SPP field. The elementary excitonic excitations of the J-aggregated thin film are superradiant J-aggregate excitons, delocalized over a few tens of molecules

[Quenzel et al, ACS Nano 16.3 (2022)]. Hence, they can be considered as independent elementary excitations, unless they are coupled to a delocalized external perturbation. In the cited work, e.g., we show the further delocalization of the J-aggregate excitons by coupling to the SPP field of a planar gold film. In the present sample, the inter-exciton coupling is provided by the delocalized and spatially inhomogeneous SPP field of the nano-slit array. As such, this coupling further delocalizes the excitonic excitation but think that it is more appropriate to understand this as an “indirect coupling”. Since the comparison between the dynamics arising from the Frenkel exciton Hamiltonian and the effective Hamiltonian in Fig. 4 shows a convincing agreement, we conclude that the assumption of two non-interacting exciton subspecies is reasonable. Therefore, we have not made any changes to the manuscript that address this point.

We wish thank all three Reviewers for their insightful comments and helping us in improving our manuscript. We hope that we have addressed all comments in sufficient detail and clarity.

REVIEWERS' COMMENTS

Reviewer #2 (Remarks to the Author):

I would like to thank the authors for addressing all my concerns. From my perspective, the manuscript is now ready for publication in Nature Communications.

Reviewer #3 (Remarks to the Author):

I appreciate the authors for diligently addressing the comments and suggestions provided for their manuscript. The authors' response, coupled with the revisions made to the manuscript, effectively and comprehensively tackled all the concerns raised. As a result, I find the manuscript well-prepared for publication in Nature Communications.

Response to the Reviewers' Reports

We thank the Reviewers again for their time and the critical evaluation of our manuscript. We are happy to hear that we were able to address all concerns and that all Reviewers are supporting publication of the revised manuscript in Nature Communications.

Reviewer #2

I would like to thank the authors for addressing all my concerns. From my perspective, the manuscript is now ready for publication in Nature Communications.

Reviewer #3

I appreciate the authors for diligently addressing the comments and suggestions provided for their manuscript. The authors' response, coupled with the revisions made to the manuscript, effectively and comprehensively tackled all the concerns raised. As a result, I find the manuscript well-prepared for publication in Nature Communications.